# Automatic Identification of Involuntary Muscle Activity in Subacute Patients with Upper Motor Neuron Lesion at Rest—A Validation Study

**DOI:** 10.3390/s23020866

**Published:** 2023-01-12

**Authors:** Andrea Merlo, Isabella Campanini

**Affiliations:** 1LAM-Motion Analysis Laboratory, S. Sebastiano Hospital, Neuromotor and Rehabilitation Department, Azienda USL-IRCCS di Reggio Emilia, Via Circondaria 29, 42015 Correggio, Italy; 2Merlo Bioengineering, 43121 Parma, Italy

**Keywords:** upper motor neuron lesion, involuntary muscle activity, automatic detection, surface EMG, sensitivity, specificity, accuracy

## Abstract

Sustained involuntary muscle activity (IMA) is a highly disabling phenomenon that arises in the acute phase of an upper motor neuron lesion (UMNL). Wearable probes for long-lasting surface EMG (sEMG) recordings have been recently recommended to detect IMA insurgence and to quantify its evolution over time, in conjunction with a complex algorithm for IMA automatic identification and classification. In this study, we computed sensitivity (*Se*), specificity (*Sp*), and overall accuracy (*Acc*) of this algorithm by comparing it with the classification provided by two expert assessors. Based on sample size estimation, 6020 10 s-long sEMG epochs were classified by both the algorithm and the assessors. Epochs were randomly extracted from long-lasting sEMG signals collected in-field from 14 biceps brachii (BB) muscles of 10 patients (5F, age range 50–71 years) hospitalized in an acute rehabilitation ward following a stroke or a post-anoxic coma and complete upper limb (UL) paralysis. Among the 14 BB muscles assessed, *Se* was 85.6% (83.6–87.4%); *Sp* was 89.7% (88.6–90.7%), and overall *Acc* was 88.5% (87.6–89.4%) and ranged between 78.6% and 98.7%. The presence of IMA was detected correctly in all patients. These results support the algorithm’s use for in-field IMA assessment based on data acquired with wearable sensors. The assessment and monitoring of IMA in acute and subacute patients with UMNL could improve the quality of care needed by triggering early treatments to lessen long-term complications.

## 1. Introduction

The onset of muscle overactivity is one of the consequences of upper motor neuron lesions (UMNL) due to strokes or traumatic brain injuries [1,2]. The initial paresis leads to lack of use of the affected limb and, if protracted over time, causes soft tissue reorganization. This, in turn, will then modify the sensitivity thresholds of the neuromuscular spindles, thus affecting the central nervous control, which eventually leads to various forms of overactivity that are often observed in neurological patients [1].

Recent literature highlights the key role of surface electromyography (sEMG) in identifying and measuring the presence or absence of overactivity [3,4,5,6,7,8,9]. Muscle overactivity can appear as early as the acute and subacute phases, even in limbs still affected by paresis [2,10]. When we clinically evaluate a muscle as hypertonic, different types of velocity-dependent overactivity may be an underlying factor, e.g., spasticity which is characterized by the absence of EMG signal at rest, and spastic dystonia that can be distinguished by the inability to silence EMG activity voluntarily [3]. This latter specific form of involuntary muscle activity (IMA) at rest has been assessed in some recent works where measurements lasted from a few seconds [11,12] up to very long-lasting acquisitions [10].

In the study by Campanini and colleagues, IMA was assessed in subacute stroke patients using high-density sEMG acquisitions that lasted five minutes [12]. Trompetto and colleagues studied the presence of IMA in chronic stroke patients using bipolar sEMG, and muscle activity at rest was found in more than 70% of the sample [11]. Finally, Merlo and colleagues have proven the possibility of assessing IMAs from the very onset by using sEMG wearable probes in a single differential configuration on acute inpatients’ biceps brachii (BB) [10]. The acquired recordings lasted up to six hours. To discard unreliable epochs, the algorithm performed a set of data quality controls on sEMG data. Next, EMG activity was detected by the presence of motor unit action potentials (MUAPs). MUAPs were identified as either separate MUAP trains or summed up as interferential signals. The information provided by an accelerometer embedded in the sensor was then used to recognize phases with muscle activity in the complete absence of movement, typical of IMA [10].

The availability of information on IMA characteristics in acute patients can be used to assess its predictive ability for the development of muscular overactivity and joint deformities in UMNL diseases [10]. These considerations could potentially have a remarkable clinical impact on an early start of rehabilitation treatments and prevent secondary complications. Nevertheless, this methodology of EMG signal processing needs validation before being used in clinical practice. In particular, the automatic classification of the algorithm must be compared with the gold standard—i.e., a manual classification performed by experienced operators.

In this study, we assessed the sensitivity and specificity of the classification procedure described in Merlo et al., 2021 [10] when used to detect IMA from in-field measurements on patients admitted to an acute rehabilitation ward.

This study aimed at validating the classification procedure described in Merlo et al., 2021 [10] when used to detect IMA from in-field measurements on patients admitted to an acute rehabilitation ward by computing its sensitivity, specificity, and overall accuracy.

## 2. Materials and Methods

### 2.1. Study Design

This is a validation study, where a test index is compared to the gold standard and follows the study on the in-field feasibility of IMA assessment by sEMG published by our group [10]. The study did not interfere at all with the clinical and rehabilitative pathways of the included patients.

It is worth highlighting that identifying and quantifying IMAs did not provide a direct clinical diagnosis. Instead, it allows for the assessment or monitoring of a pathophysiological condition.

### 2.2. Definitions

In this study, we shall use the following definitions, according to Merlo and colleagues 2021 [10]:sEMG activity: any presence of muscle activity during a recording, irrespective of the amount of upper limb (UL) acceleration recorded by the probe—i.e., with the patient at rest, during passive mobilization, during posture changes (e.g., from supine to sitting), and also during nursing or physiotherapy activities.IMA: the presence of muscle activity without any UL acceleration, with the patient at rest.Motion Artifact (MoA): any signal variation due to a skin-electrode interface change, e.g., during patient handling and physiotherapy treatments.EMIs: interferences generated by motors and pumps positioned close to the patient, by capacitive couplings during patient handling, or due to loss of adhesion in one or both electrodes.

### 2.3. Protocol and Data Acquisition Procedures

Data acquisition was performed by wearable probes (Mini Wave Plus, Cometa, Italy) for the acquisition of bipolar sEMG data and 3D acceleration at a sampling frequency of 2 kHz. These probes are fitted with onboard memory, where data can be stored continuously for several hours.

Circular disposable Ag/AgCl electrodes with a hydrogel conductive paste, and a round foam edge (Arbo-Kendall H124, with a 16 mm diameter contact area), were placed on the BB according to SENIAM recommendations [13,14], with a center-to-center inter-electrode distance of 25 mm. These dimensions implied spatial filtering that was not critical for this application [15].

The probe was fixed to the skin with double-sided adhesive patches (provided by the manufacturer). The probe was then activated using a remote control, and it was left in place for approximately six hours between 8:00 a.m. and 2:00 p.m. During data acquisition, the in-ward daily activities and routines remained the same. These included medications, nursing, rehabilitation activities, meals, resting, or sleeping.

At the end of the recordings, data were downloaded to a PC and split into 5 min time segments, which meant approximately 70 files per muscle per subject.

A complete description of the acquisition protocol can be found in Merlo et al., 2021 [10].

### 2.4. Patient Characteristics

The study included data from 10 patients, five males and five females, aged between 50 and 71 (see Table 1). Included patients were hospitalized in the acute rehabilitation ward at our Institution having suffered either an ischemic stroke (3 cases), a hemorrhagic stroke (4 cases), or a post-anoxic coma (3 cases), and received routine clinical care. Time elapsed from the acute event ranged between 18 and 132 days.

All patients had complete UL paralysis (absence of any voluntary movement from the shoulder to the fingers). This ensured that only IMA and no other types of overactivity were recorded.

Four patients had bilateral UL plegia, for a total of 14 analyzed BBs.

The local Ethical Committee approved the study (2015/0015247). Informed consent was obtained prior to enrollment either from the patients themselves or from their legal guardians, as per Italian law.

### 2.5. Size Design for Sensitivity and Specificity Assessment 

This study aims to determine the sensitivity and specificity of the algorithm proposed by Merlo et al. to detect IMA in acute patients with UMNL [10].

To reach our target, the number of epochs to be analyzed was computed considering this prevalence and below equations [16]:(1)$$n_{Se}=\frac{Z_{\frac{\alpha }{2}}^{2}\hat{Se}\left(1-\hat{Se}\right)\;}{d^{2}\times Prev\;}$$
(2)$$n_{Sp}=\frac{Z_{\frac{\alpha }{2}}^{2}\hat{Sp}\left(1-\hat{Sp}\right)\;}{d^{2}\times \left(1-Prev\right)}$$
where $\hat{Se}$ and $\hat{Sp}$ are the expected sensitivity and specificity, respectively, *d* is the desired half-width of the confidence interval, and *Prev* is the prevalence of the phenomenon. 

For a = 0.05, $Z_{\alpha /2}=1.96$ is inserted. The results of the study by Merlo and colleagues suggest a 25% prevalence of IMA among epochs and subjects. When setting both $\hat{Se}$ and $\hat{Sp}$ to 75% and the width of the confidence interval to 5% (i.e., d = 2.5%), a minimum of 4610 epochs was obtained. A narrow confidence interval was selected to obtain a reliable estimation of both $\hat{Se}$ and $\hat{Sp}$.

To account for the epochs to be excluded due to arm movement (e.g., during nursing activities or physiotherapy), a 20% drop-out rate has been considered [10], resulting in a minimum of 4610/0.80 = 5763 epochs to be collected and analyzed.

In this study, we compared the index test and the reference standard on 6020 epochs. These were 430 randomly selected epochs from all acquisitions available for the 14 BB muscles (14 × 430 = 6020).

### 2.6. Index Test

The algorithm published in Merlo et al., 2021 [10] analyzes consecutive epochs lasting 10 s.

Each epoch undergoes a set of quality controls in both the time and the frequency domains to identify and discard epochs corrupted by either motion artifacts or electromagnetic interference. More specifically, each epoch is classified as containing unreliable data if at least one of the following exclusion conditions is met:number of EMI-related harmonics ≥ 5, in the frequency domain;epoch peak-to-peak amplitude range > 2000 mV, in the time domain;epoch peak-to-peak amplitude range < 50 μV, in the time domain;epoch minimum spectral peak frequency < 25 Hz, in the frequency domain;epoch minimum spectral mean frequency < 30 Hz, in the frequency domain.

The presence of muscle activity was later assessed using the method described by Merlo et al., 2021 [10]. It uses a wavelet-based enhancement of the matched filter technique to identify the presence of each MUAP-like shape in the signal. The presence of a rhythmic sequence of MUAPs with a minimum physiological firing rate was later verified. Only when this condition was met was the signal segment containing rhythmic or interferential MUAPs classified as muscle activity.

The 3D accelerometric sensor embedded in the probe was used to detect and ignore the epochs with any arm movement, according to the definition of IMA. For each 10 s epoch, the presence of arm movement (yes/no) was detected when the variation in the modulus of the acceleration signal (i.e., the magnitude of the vector addition of the three components) was higher than the set threshold of 30 × 10^−3^ g. This threshold was set, based on the study by Hurter and colleagues [10,17]. Under static conditions, the modulus is equal to the force of gravity. Acceleration variations are expected during passive movement due to external interventions (e.g., nursing activities, physiotherapy).

The presence of muscle activity without any arm movement was classified as IMA.

### 2.7. Reference Standard

Two assessors (authors AM and IC) visually inspected all epochs. Both have over two decades of experience with surface EMG and its use in neurorehabilitation including many publications surrounding the topic [18,19,20,21,22,23,24].

Since the number of epochs to be rated was very high, custom software was implemented to maximize the ergonomics of the assessment and lessen the assessors’ workload, including potential errors due to fatigue. In brief, the software displayed the epoch to be assessed and paused. All lower left keyboard keys could be used to indicate “No” (i.e., no EMG) and all upper right keys could be used to indicate “Yes” (i.e., EMG detected). In this way, the assessor’s gaze remained fixed on the screen, and a slight movement of their left or right hand was enough to rate each epoch. A new epoch was automatically displayed after each assessment. All epochs were prepared beforehand and saved as pictures to eliminate the time delay required by data reading. Special keys were also available, e.g., the backslash to move back to the previous epoch (in case of error), the asterisk to add a comment, and the escape key to exit. The assessment could be stopped and restarted at any given time.

All epochs were evaluated by both experts completely independently. In case of a disagreement, epochs were *a posteriori* reviewed jointly by both assessors and re-evaluated considering both the raw signal and its time-frequency transform, until an agreement was reached.

### 2.8. Statistical Analysis

The performance of the algorithm proposed in Merlo et al., 2021 [10] to automatically detect IMA from in-field sEMG measurements was determined by comparing it to the classification provided by expert assessors in a two-way table, and by computing its sensitivity and specificity along with their respective 95% confidence intervals (CI). The expert assessors (authors AM and IC) have >20 years of clinical and research experience with sEMG and its use with neurologic patients.

Finally, the false-positive (FP) and false-negative (FN) epochs and their characteristics were also analyzed and discussed.

### 2.9. Study Reporting

This manuscript has been written according to the Reporting Diagnostic Accuracy Studies standards (STARD guidelines) [25].

## 3. Results

Out of 6020 epochs, a movement was detected by the accelerometer in 1324, for a total of 4696 remaining epochs in which IMA might have been present. The two-way table comparing the algorithm’s classification to the reference standard on these 4696 epochs is reported in Table 2.

*Se* was 85.6% (83.6–87.4%) and *Sp* was 89.7% (88.6–90.7%). The overall accuracy of the algorithm in detecting IMA was 88.5% (87.6–89.4%).

A few examples of correctly and incorrectly classified epochs is shown in Figure 1.

The distribution of the RMS amplitude in the four subsets (TP, FP, TN, FN) is presented in Figure 2. RMS amplitude data were trimmed to their 99th percentile—i.e., rare high-amplitude artifacts were excluded, in order to use the same full-scale for all plots. On the one hand, some differences can be appreciated among subjects (e.g., 001-B, 002-E, 008-X). On the other hand, the similarity among TP, FP, and TN distributions supports the selection of an algorithm based on the recognition of the MUAP shape, as used in this study.

TP epochs contained either a short train of MUAPs or an interferential signal, as in Figure 1a,b. The median RMS amplitude of the true positive epochs, (i.e., of IMA) was 12 mV, and it ranged between 2 and 92 μV. The median peak-to-peak amplitude for these epochs was, on average, 120 mV and it ranged between 24 and 1400 μV. The combination of low RMS values with high peak-to-peak values originated from trains of a single or a few motor units, without amplitude summation, as in Figure 1a. The average duration of the onset period detected with the 10 s epochs was 9.63 (1.95) seconds.

The number of FP epochs per subject varied between 2 and 65. Most FP epochs consisted of a very low amplitude baseline, with some amplitude modulation that resulted in the false detection of a short EMG burst, as shown in Figure 1c. The average duration of these onset periods was 5.43 (4.99) seconds. There were cases where a high-frequency heartbeat was misclassified as a train of MUAPs (Figure 1d). The remaining FPs resulted from higher amplitude modulated EMIs that were not detected by the first stages of the algorithm.

The number of FN epochs per subject varied between 1 and 35. Most FN epochs (120/195) were discarded by the algorithm because of a preliminary control on the average amount of EMIs in the whole file (5 min long recording), even if the specific epoch was not corrupted. This may have happened when a medical machine placed close by was activated during the 5 min recording. The remaining FN epochs typically contained either isolated MUAPs (Figure 1g) or sEMG corrupted by large motion artifacts (Figure 1h). 

TN epochs could contain a baseline noise only or a baseline noise with some ECG (Figure 1e), EMIs (Figure 1f), motion artifacts, or ECG.

### 3.1. Accuracy at the Single-Subject Level

Among the 14 BB muscles assessed in the ten-patient sample, the overall accuracy ranged between 78.6% and 98.7%, as presented in Table 3. *Acc* was lower than 80% in two cases, ranged between 80% and 90% in six cases, and was greater than 90% in five cases. No activity was found in one of the assessed muscles (ID 00J-Vl); hence, Acc was not computable.

### 3.2. Level of Agreement between Assessors

The two assessors agreed 91.4% of the time. This confirms the feasibility of the visual evaluation of sEMG epochs. Typically, the assessors did not agree when MUAPs could barely be distinguished from baseline and the signal had a very low amplitude. At this stage, a wavelet-based time-frequency analysis was used to analyze epochs and reach a consensus. This approach allowed the assessors to clearly identify low-amplitude modulated EMIs. Figure 3 shows an example of this analysis.

## 4. Discussion

In this study, we tested the performance of an algorithm for IMA automatic identification and classification to be used with long-lasting sEMG acquisitions in neurological patients at their bedside.

The study results provided the values of *Se* and *Sp*, along with the CI for the new algorithm. Both ratios reached the 90% value, with an overall *Acc* of 88.5%, revealing that the index test is feasible and can be used for research purposes.

In the literature, algorithm accuracy for automatic classification of bioelectric signals is higher than the one in our study when this detection is related to critical conditions in the patient’s life. The detection of heart failure, pacing artifacts, or noise with electrocardiography (ECG) is performed with algorithms whose accuracy ranges from 97% to 100% [26,27,28]. Conversely, algorithms used in non-critical applications have an accuracy similar to that of the algorithm analyzed in this study. This happens in instances of sleep arousal detection, schizophrenia diagnosis with electroencephalography (EEG), or when evoked potentials are applied to detect the onset of coma, or deafness degree [29,30,31,32]. The accuracy of these algorithms ranges from 82% to 93%. Such slightly lower values are justifiable since those measurements are not taken in critical conditions, such as for IMA detection described in the current study.

When considering patient analysis individually as shown in Table 3, *Acc* ranged from 78.6% to 98.7%. This variability was mainly due to the patient’s data characteristics. The worst result was characterized by signals with a very low signal-to-noise ratio, i.e., epochs where the sEMG minimally emerged from the baseline. The best results were obtained from mid-amplitude MUAPs or interferential signals and a very low baseline RMS amplitude (i.e., a high signal-to-noise ratio). 

Depending on the aim of the research (i.e., early identification of IMA occurrence in acute patients versus monitoring of IMA evolution in subacute or chronic patients), researchers may prefer higher levels of sensitivity or specificity.

The ease of obtaining long-lasting data by wearable probes and its in-field feasibility supports the adoption of the presented technique already in the acute phase. A biomarker based on the presence of acute IMA could be predictive of the long-term existence and severity of muscle overactivity (e.g., spasticity) typical of patients with stroke outcomes.

Besides the performance at the single epoch level, it is worth noting that all patients with IMA have been adequately identified. Whether the primary clinical question was the presence or absence of IMA, the proposed methodology for measuring and monitoring IMA using wearable sEMG probes has proven to be a reliable tool. 

In all previous studies, the presence of IMA was assessed by visual inspection of sEMG traces lasting only a few minutes [33,34]. The algorithm verified in this study can detect a steady presence of MUAPs, including the advantage of working automatically. In the literature, automated recognition techniques of specific alterations in muscle behavior (clones and spasms) caused by lesions in the central and peripheral nervous system have been successfully implemented, using long-lasting sEMG acquisitions, thus supporting this approach in the clinical field [35,36]. 

The presented approach could be used to monitor and quantify overactivity in patients with upper motor neuron lesions of different etiology, such as traumatic brain injury and cerebral palsy, and in patients with spinal cord injury. 

### Limitations and Future Developments

The dataset we used to validate the study is the same as in Merlo et al., 2021 [10]. Although this dataset is extremely large, with 154,440 available epochs, all data come from a single ward at the same institution. This might limit the external validity of our results. It is therefore advisable to verify the algorithm’s performance on data acquired from different patients and in other locations, where IMA characteristics and the type of recorded EMIs could differ.

The accuracy of the algorithm could be improved when analyzing the amplitude of FPs and FNs (See Figure 2) and their duration. A threshold on the minimum duration of the detected activation could be set within a single epoch (e.g., minimum activation time > 5 s). To further improve its robustness, a reasonable threshold in the number of consecutive epochs containing IMA could be added. This is in line with the pathophysiology of IMA, which is characterized by an impaired derecruitment ability. 

A way to reduce false negatives could be achieved by reducing the time window for the initial quality control on the number of power line harmonics from 5 min to 30 s or 60 s. This would avoid discarding epochs belonging to files only partially corrupted by EMIs produced by pumps or motors close to the patients. Better algorithm performances could be obtained by customizing the onset to the patient’s baseline noise characteristics.

## 5. Conclusions

Sensitivity and specificity of the algorithm to detect IMA from sEMG in-field measures in patients admitted to an acute rehabilitation ward are adequate for its use in research studies. The presence of IMA was detected correctly in all patients. 

The assessment and monitoring of IMA in acute and subacute patients with UMNL by using wearable sensors could improve the quality of the care provided by promoting early treatment and lessen potential long-term complications.

## Figures and Tables

**Figure 1 sensors-23-00866-f001:**
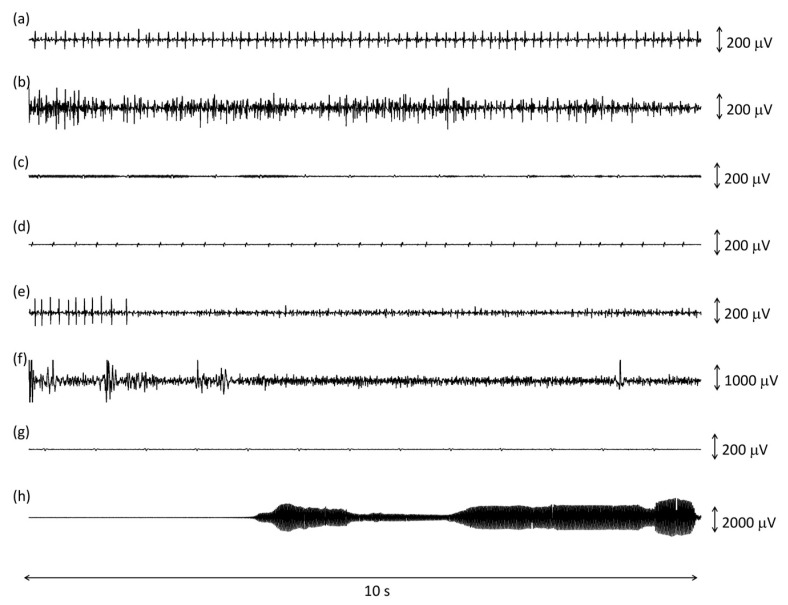
Examples of 10 s epochs analyzed by the algorithm classified as true positive (**a**,**b**), false positive (**c**,**d**), false negative (**e**,**f**), and true negative (**g**,**h**) when compared to the golden standard. See text for further details.

**Figure 2 sensors-23-00866-f002:**
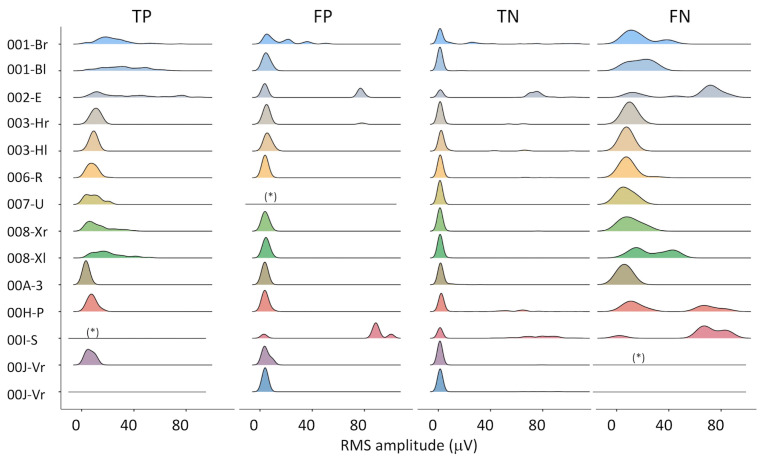
Distribution of the RMS amplitude for all epochs of all subjects, for true positive (TP), false positive (FP), true negative (TN), and false negative (FN) classifications. Legend: (*) less than 3 epochs in this class, distribution not drawn; flat lines: no epochs in this class.

**Figure 3 sensors-23-00866-f003:**
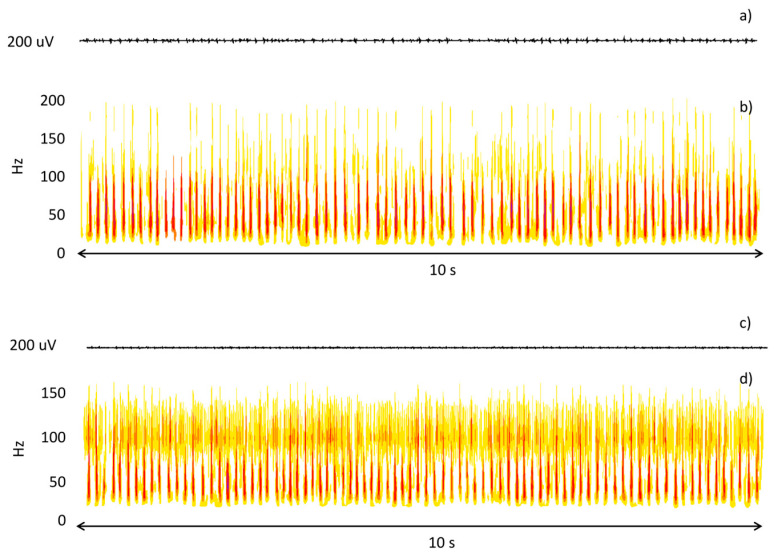
Time (**a**,**c**) and time-frequency (**b**,**d**) representation of two epochs subject to double check due to a disagreement between the expert assessors. In the first case, the time-frequency analysis highlights the presence of MUAPs; in the second case, it highlights the presence of EMIs.

**Table 1 sensors-23-00866-t001:** Sample Characteristics.

ID	Sex	Age (Years)	Etiology	Days from Lesion	Affected Limb
001-B	F	69.3	Post-anoxic coma	132	R, L
002-E	F	53.1	Hemorrhagic stroke	50	R
003-H	F	59.1	Post-anoxic coma	109	R, L
006-R	M	63.4	Hemorrhagic stroke	18	R
007-U	M	66.2	Ischemic stroke	53	R
008-X	M	70.9	Ischemic cerebellar stroke	59	R, L
00A-3	F	56.4	Hemorrhagic stroke	28	R
00H-P	F	60.8	Hemorrhagic stroke	120	L
00I-S	M	49.4	Ischemic stroke	61	L
00J-V	M	62.5	Post-anoxic coma	28	R, L

**Table 2 sensors-23-00866-t002:** Two-way table of the algorithm-based and the gold-standard classification.

	Gold Positive	Gold Negative	Total
Tested Positive	1158	345	1503
Tested Negative	195	2998	3193
Total	1353	3343	4696

**Table 3 sensors-23-00866-t003:** Patient ID, recorded hours, and IMA detection accuracy (95%CI) for all assessed biceps brachii muscles. Both arms were assessed in bilateral patients, as indicated by the ‘l’ and ‘r’ letters in the ID. A two-way table was created for each muscle assessed comparing the algorithm-based and the gold-standard classification. The classification accuracy (TP + TN/Total) and its 95% confidence interval were computed and reported.

ID	Recorded Data (Hours)	Accuracy (95%CI)
001-Br	35.2	86.5% (82.8%–90.1%)
001-Bl	35.2	90.9% (87.8%–93.9%)
002-E	59.9	86.1% (82.5%–89.8%)
003-Hr	5.0	86.3% (82.9%–89.7%)
003-Hl	5.0	78.6% (74.6%–82.7%)
006-R	61.8	81.2% (76.6%–85.8%)
007-U	8.0	98.7% (97.7%–99.8%)
008-Xr	46.8	84.4% (80.4%–88.4%)
008-Xl	46.8	78.8% (74.1%–83.5%)
00A-3	12.7	92.0% (88.8%–95.2%)
00H-P	30.1	84.4% (80.3%–88.4%)
00I-S	59.6	94.3% (91.6%–97.0%)
00J-Vr	11.7	98.6% (97.5%–99.8%)
00J-Vl	11.7	N.A.

Legend: N.A. not applicable since no activity was found by the expert assessors (i.e., TP = 0).

## Data Availability

The data presented in this study are available on request from the corresponding author.

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
