# Peer review of "Automatic Identification of Involuntary Muscle Activity in Subacute Patients with Upper Motor Neuron Lesion at Rest—A Validation Study"

_sensors, 2023, doi:10.3390/s23020866_

Round 1

Reviewer 1 Report

The article deals with the problem related to the involuntary muscle activity of the upper limbs, which occurs in the presence of an upper motor neuron lesion (UMNL). Currently, in clinical practice, it is difficult to identify this disorder as early as possible. Early detection of this disorder would allow medical professionals to begin appropriate treatment or rehabilitation sooner. Recording of muscle surface EMG signals is one of the most accurate methods, but the interpretation of the data is difficult, especially for people without experience in this field. Automatic detection and assignment of EMG signal patterns to a certain disorder would allow such an algorithm to be successfully used in clinical practice.

The research work is very interesting and valuable, but I have some questions and comments:

1.       I recommend the authors specify their purpose more clearly, providing the criteria by which validation will be achieved (introduction part line 71).

1.       Methods part lacking important details:

·         The symbols (Z, α) in the formulas in subsection 2.3 should be explained in more detail. The choice of such Se, Sp and d values should be further explained and justified (lines 139-151).

·         It was also unclear whether the distribution of the data was checked before reporting them as mean (SD) (Figure 2, line 242, table 2).

·         The presentation of the accuracy data in Table 2 should be clarified. Is this mean (CI)?

2.       You claim that the accelerometer measures acceleration in all three axes (line 178). However, it remains unclear, is the same threshold value set for all axes? Also, how was the component of gravity appearing on the vertical axis evaluated? The methodological part should be significantly supplemented.

3.       Could your proposed algorithm also be used for lower limbs, for example, for cerebral palsy? Perhaps the applicability of this algorithm can be extended?

Minor comments:

·         The equations on page 3 are missing numeration.

·         I would suggest you put the grids and extend the y-axis limits to a value of 30 in Figure 2.

Reviewer 2 Report

This paper’s manuscript is fulfilled, but the content is not properly clear proposed. For high quality publication consideration, many improvements must be provided. Especially, the results section does not provide the exact results of this domain to readers. Significant improvement of result content writing must be provided for publication in my opinions.

Specific comments are as follows:

(1)    In section 2.7. Study reporting shows the manuscript follows STARD guidelines, but I can't see any flow diagram or participants' table. It's might improve reading.

(2)    In section 3.1 Accuracy at the single-subject level, "Acc was 80% in two cases" is missing the word "below" between "was" and "80%". Also, one of the case Acc is "N.A." must be described more detail in this section.

(3)    Please define the “expert assessors” who are they, such as background, ... Also, I think the definition should be more clearly, in order to reduce expert decision bias.

(4)    Personally, I would like to see more details about how this algorithm detect IMA.

Round 2

Reviewer 1 Report

Thanks to the authors for considering the recommendations I offered.